# Loss of the Mitochondrial Fission GTPase Drp1 Contributes to Neurodegeneration in a *Drosophila* Model of Hereditary Spastic Paraplegia

**DOI:** 10.3390/brainsci10090646

**Published:** 2020-09-17

**Authors:** Philippa C. Fowler, Dwayne J. Byrne, Craig Blackstone, Niamh C. O’Sullivan

**Affiliations:** 1UCD School of Biomolecular and Biomedical Sciences, UCD Conway Institute, University College Dublin, Dublin 4, Ireland; philippa.fowler@ucdconnect.ie (P.C.F.); dwayne.byrne@ucdconnect.ie (D.J.B.); 2Cell Biology Section, Neurogenetics Branch, National Institute of Neurological Disorders and Stroke, National Institutes of Health, Bethesda, MD 20892, USA; blackstc@ninds.nih.gov

**Keywords:** endoplasmic reticulum, mitochondria, fission, neurodegeneration, autophagy, *Drosophila*

## Abstract

Mitochondrial morphology, distribution and function are maintained by the opposing forces of mitochondrial fission and fusion, the perturbation of which gives rise to several neurodegenerative disorders. The large guanosine triphosphate (GTP)ase dynamin-related protein 1 (Drp1) is a critical regulator of mitochondrial fission by mediating membrane scission, often at points of mitochondrial constriction at endoplasmic reticulum (ER)-mitochondrial contacts. Hereditary spastic paraplegia (HSP) subtype SPG61 is a rare neurodegenerative disorder caused by mutations in the ER-shaping protein Arl6IP1. We have previously reported defects in both the ER and mitochondrial networks in a *Drosophila* model of SPG61. In this study, we report that knockdown of Arl6IP1 lowers Drp1 protein levels, resulting in reduced ER–mitochondrial contacts and impaired mitochondrial load at the distal ends of long motor neurons. Increasing mitochondrial fission, by overexpression of wild-type Drp1 but not a dominant negative Drp1, increases ER–mitochondrial contacts, restores mitochondrial load within axons and partially rescues locomotor deficits. Arl6IP1 knockdown *Drosophila* also demonstrate impaired autophagic flux and an accumulation of ubiquitinated proteins, which occur independent of Drp1-mediated mitochondrial fission defects. Together, these findings provide evidence that impaired mitochondrial fission contributes to neurodegeneration in this in vivo model of HSP.

## 1. Introduction

Mitochondria are highly dynamic organelles, constantly changing their size, shape, number, and location throughout cells through the opposing forces of fission and fusion [1]. These mitochondrial processes are tightly regulated to maintain mitochondrial health, which is particularly important in cells with high energy demands such as neurons. Mitochondrial fission contributes to quality control by enabling the removal of damaged mitochondria. Defects in mitochondrial fission result in a hyperfused network consisting of highly elongated and inter-connected mitochondria [1,2]. The guanosine triphosphate (GTP)ase dynamin-related protein (Drp1) is the integral facilitator of mitochondrial fission. Drp1 is recruited to form a multimeric ring-like structure around the mitochondrial outer membrane, which undergoes GTP hydrolysis to constrict the mitochondrial membrane leading to a scission event [3,4]. Mitochondrial fission mediated by Drp1 is vital during neuronal differentiation and development [5,6]. Similarly, in adult neuronal axons, tight regulation of mitochondrial fission by Drp1 is essential: loss of Drp1 is found to compromise mitochondrial bioenergetics and synaptic function [7], while a role for Drp1 has been suggested in several neurodegenerative diseases including Alzheimer’s disease [8,9], Parkinson’s disease [10] and amyotrophic lateral sclerosis (ALS) [11].

Coordination of Drp1-mediated mitochondrial fission also involves the endoplasmic reticulum (ER). ER tubules form contacts marking sites for mitochondrial fission [12] while the ER-localised inverted formin 2 (IFN2) mediates the initial mitochondrial constriction via polymerisation of actin filaments [13]. Furthermore, the ER functions as a platform for Drp1 oligomerisation before transfer to mitochondria by the Drp1 receptors Mff, Fis1, MiD49 and MiD51 to contribute to fission [14]. Dynamic post-translational modification of Drp1 alters its ability to promote mitochondrial fission and through its functions, the ER additionally contributes to the regulation of these modifications. Increased cytosolic calcium promotes activation of Drp1 by dephosphorylation at S637 [15,16], while the ER resident protein disulfide isomerase (PDI) has been shown to alter S-nitrosylation and phosphorylation of Drp1 to modify mitochondrial morphology in neurons [17].

Hereditary spastic paraplegias (HSPs) are a genetically heterogeneous group of disorders characterised by degeneration of the longest motor neurons in the corticospinal tract, resulting in lower limb spasticity. The most common mutations causing HSP are loss-of-function mutations in genes encoding ER-shaping proteins which function to regulate the organisation of the ER network [18]. Given the important role of the ER in the regulation of mitochondrial fission, it has been hypothesised that dysregulation of this process may contribute to neurodegeneration in HSP. Animal models of HSP generated by loss of the ER-shaping proteins Atlastin, Reticulon or ARL6IP1 result in mitochondrial elongation and defective mitochondrial fission in neurons [19,20]. Fibroblasts from HSP patients with mutations in the ER-shaping protein REEP1 also display elongated mitochondria and hyperphosphorylated Drp1 compared to healthy controls which can be rescued by overexpression of Drp1 or inhibiting Drp1 S637 phosphorylation [21]. Understanding the role of Drp1-mediated mitochondrial fission in HSP-associated neurodegeneration is important as it may serve as a potential therapeutic target for some forms of these disorders.

In this study, we set out to investigate the role of mitochondrial fission in mediating neurodegeneration in a *Drosophila* model of HSP generated by targeted knockdown of the ER-shaping protein Arl6IP1. We show that reduction in Arl6IP1 in vivo decreases levels of Drp1, reduces ER–mitochondrial contacts and increases mitochondrial elongation in motor neuron axons. Increasing mitochondrial fission by overexpression of Drp1^WT^, but not dominant negative Drp1^K38A^, rescues axonal mitochondrial morphology and partially restores the progressive locomotor defects caused by knockdown of Arl6IP1. However, not all defective processes in *Arl6IP1* RNAi flies are restored by overexpression of Drp1. We find evidence of disrupted protein processing in *Arl6IP1* RNAi flies, specifically impaired induction of autophagy and increased aggregation of ubiquitinated proteins, which occurs independent of Drp1-mediated mitochondrial fission defects.

## 2. Materials and Methods

### 2.1. Drosophila Stocks and Conditions

Flies were raised on standard yeast, dextrose, cornmeal and agar food at 25 °C with a 12:12 light-dark cycle and transferred to fresh vials every 2–3 days. For Arl6IP1 knockdown experiments, the fly lines used were *w*^1118^ GD and KK control stocks 60,000 and 60,100 and *UAS-Al6IP1-RNAi* GD and KK (line numbers 5894 and 10,790, respectively), all obtained from the Vienna *Drosophila* RNAi Centre, www.vdrc.at [22]. Both libraries were constructed by inserting inverted repeats of known genes into *Drosophila*, with the key difference being that GD library insertions are P-element based transgenes with random insertion sites, whereas the KK library contains phiC31-based transgenes with a single, defined insertion site. Phenotypes were analysed in RNAi knockdown lines compared to their appropriate controls. To study Drp1, the fly lines used were *UAS-Drp1* provided by Mel Feany (Harvard Medical School) [23], *Drp1:: FLAG-FlAsH-HA* obtained from the Bloomington Stock Centre [24] and *UAS-Drp1^wt^HA* and *UAS-Drp1^K^*^38*A*^*HA* provided by Prof. Jongkyeong Chung (Korea Advanced Institute of Science and Technology) [25]. UAS lines were crossed to *da-GAL4* [26] for the generation of samples used for sqPCR, *nSyb-GAL4* [27] for the generation of crosses used for behavioural studies, *OK6-GAL4* [28] for mitochondrial analysis and *Cg-GAL4* [29] for larval fat body analysis. Other fly stocks used were mito::GFP (stock number 8443) [30] and the autophagosomal marker GFP-mCherry-Atg8a (stock number 37,749) [31] both obtained from the Bloomington Stock Centre. 

### 2.2. Semi-Quantitative PCR

Total RNA was purified from ten third instar stage larvae (L3) via treatment with TRIzol reagent (Invitrogen, Paisley, UK) as previously described [19]. Primers used were: Rp49-F: 5′-CCGACCACGTTACAAGAACTCTC-3′; Rp49-R: 5′-CGCTTCAAGGGA CAGTATCTGA-3′; Arl6IP1-F: 5′-GGTGCTGTGGTACCTGGACT-3′; Arl6IP1-R: 5′-CCAT AGCCAAAAGACCCAAA-3′; Drp1-F: 5′-ACTGGTGCTCCAGCTGATCT-3′; Drp1-R: 5′- GATGGAGTTCGGGTTCTCAA-3′. PCR conditions were: 94 °C for 45 s, 60 °C for 45 s and 72 °C for 1 min, repeated for 22 cycles for *Rp49*, 26 cycles for *Arl6IP1* and 29 cycles for *Drp1*. Following PCR, products were visualized via a gel containing ethidium bromide.

### 2.3. Immunoblot Analysis

Soluble protein lysate was purified from 20 adult *Drosophila* as described previously [32]. Samples were run on 10% polyacrylamide gels and electrophoretically transferred to nitrocellulose membranes (0.45µm, GE Healthcare Sciences, Amersham, UK). Nitrocellulose membranes were then blocked in 5% milk and incubated with primary antibodies against the HA-epitope tag (MMS-101P-100, Eurogentec; Liege, Belgium) and tubulin (T9026, Sigma; Dorset, UK). Bands were subsequently visualised using anti-mouse IgG antibodies conjugated with IR800 fluorescence dye (#SAS-35521, Thermo Fisher Scientific; Waltham, MA, USA). Membranes were imaged using the LI-COR Odyssey Infrared Imaging System, and intensities were determined using ImageJ 1.48v.

### 2.4. Histology and Immunomicroscopy

Immunostaining of *Drosophila* larvae was carried out as described previously [19]. Briefly, L3, or late L2 for analysis of autophagic flux, larvae were dissected in chilled in Ca^+2^-free HL3 solution [33], fixed in 4% formaldehyde in PBS for 30 min, permeabilised in 0.1% Triton X-100 in PBS and incubated with primary antibodies against HA (as above), Dlg (4F3, Hybridoma; Iowa City, IA, USA), GFP (A-6455, Thermo Fisher Scientific; USA), HRP (P7899, Sigma) and FK2 (BML-PW8810, Enzo Life Sciences; Farmingdale, NY, USA). Following incubation with the appropriate secondary antibodies, fixed preparations were mounted in Vectashield containing the nuclear stain DAPI (Vector Laboratories; Oxfordshire, UK) and were viewed using an Olympus IX81 confocal head mounted on an Olympus Fluoview FV-1000 microscope. All images were acquired using a 60×/1.35 NA objective and FV10-ASW cer.04.01 software. All axonal analyses were conducted by imaging motor axon bundles passing through segment A7. Similarly, analyses of neuromuscular junctions (NMJs) were conducted by imaging muscles 6 and 7 at segment A7. *Z*-stacks of larval fat bodies were imaged at approximately 2-μm intervals and were subsequently converted into maximum intensity projections for image analysis. 

### 2.5. Image Analysis

All image analysis was carried out blind to genotype.

#### 2.5.1. Drp1 Localisation Analysis

For Drp1 localisation studies, mitochondrial fission and post fission sites were quantified by measuring the number of interactions between mitochondria and Drp1::HA, where mitochondrial fission sites were defined by Drp1 staining at points of decreased mito::GFP staining and post-fission sites were defined by Drp1 staining at the ends of mito::GFP tubules. 

#### 2.5.2. Mitochondrial Analysis

Mitochondrial staining at NMJs was quantified in ImageJ. Mean grey intensity values of mito::GFP within terminal synaptic boutons were normalised to bouton area as defined by the post-synaptic protein Dlg staining. For mitochondrial circularity, images were first thresholded and converted to binary images in ImageJ. The watershed function was used on binarized images to separate adjoined mitochondria. Mitochondrial circularity was quantified using the Shape Descriptors option in the ImageJ/Analyse menu which uses the equation 4π × [Area]/[Perimeter] 2 to generate a value between 0 and 1, where 1 is a perfect circle and as the value approaches 0 it becomes increasingly elongated.

#### 2.5.3. Autophagic Flux Analysis

Co-localisation of Atg8-mCherry and GFP puncta was quantified manually using the threshold function in ImageJ. For this, the channel corresponding to mCherry was thresholded to highlight individual puncta and converted into a binary image; the watershed function was used to separate adjoining puncta. Using this as a mask, the mean grey values of puncta were subsequently quantified in channels corresponding to mCherry and αGFP. Red: green (mCherry:αGFP) ratios were calculated by dividing the mean grey values in the red channel by the green channel, where an increase represents a drop in αGFP signal and therefore an increased formation of autolysosomes.

#### 2.5.4. Quantification of Ubiquitinylated Proteins

To quantify Fk2 staining within terminal synaptic boutons, mean grey intensity values were quantified and normalised to bouton area using HRP as a marker for neuronal membranes.

### 2.6. Live Mitochondrial Trafficking in Axons

To examine axonal mitochondrial trafficking, L3 stage larvae were dissected in HL3 supplemented with 1 mM Ca^2+^ and 4 mM L-glut pre-warmed to 25 °C. Images were acquired using a Zeiss AxioImager M1 upright fluorescent microscope with a 63×/0.95 NA objective water dipping lens every 2 s for 2 min to generate a movie. Dissected larvae were imaged no later than 30 min from the time of dissection. To determine average mitochondrial speed, at least 3 kymographs were generated per movie using the multi-kymograph plugin in Image J. Individual mitochondrial velocities were determined from the kymographs using the segmented line tool to trace mitochondrial distance over time, and average anterograde/retrograde speed was calculated using the read velocities from tsp macro. The proportion of mobile mitochondria was determined by manual categorisation of the movement of mitochondria within randomly placed 10 µm^2^ boxes, with an average of 2–3 boxes analysed per movie. 

### 2.7. Electron Microscopy

Early L2 larvae, which could be accommodated by our instrumentation, were subjected to high pressure freezing using a BAL-TEC HPM010 freezer (ABRA Fluid AG; Widnau, Switzerland). Frozen larvae were transferred into cryovials containing a pre-cooled mix of 1% osmium tetroxide, 0.1% uranyl acetate in 95% acetone/5% H_2_O. Specimens subsequently underwent freeze substitution for 48 h at −80 °C, then were transferred to −20 °C overnight, removed and allowed to reach room temperature for ~4 h. Following a rinse in pure acetone, specimens underwent propylene oxide then propylene oxide/epoxy resin infiltration, before being embedded in 100% resin. Ultrathin sections (80 nm) were obtained using a diamond knife and microtome, collected on copper grids and subsequently post-stained with uranyl acetate and lead citrate. Sections were imaged at 120 kV using a FEI Technai 12 transmission electron microscope (EM) and at captured magnifications of 9000× and 25,000×. EM images were collected from 3–4 larvae per genotype, with a minimum of seven epidermal cells per larvae. 

### 2.8. Ultrastructural Analysis

For ER and mitochondrial analyses, the free hand tool in ImageJ was used to manually trace and measure ER and mitochondrial branch length. For ER–mitochondrial contact analyses, the circumference of each mitochondrion and the proportions of mitochondrial surface closely opposed to the ER (<30 nm) were calculated. The number and length of individual ER–mitochondrial contacts were similarly calculated. The line tool in ImageJ was used to measure the distance between the ER and mitochondria to determine contact point thickness. All image analyses were carried out blind to genotype. 

### 2.9. Behavioural Analysis

All behavioural assays were conducted on are progeny of *nSyb-GAL4* flies crossed to appropriate *w*^1118^ control stocks: UAS-*Arl6IP1* RNAi (*Arl6IP1* RNAi), UAS-*Arl6IP1* RNAi*.UAS-Drp1^WT^* (*Arl6IP1* RNAi + *Drp1^WT^*) or UAS-*Arl6IP1* RNAi*.UAS-Drp1^K^*^38*A*^ (*Arl6IP1* RNAi + *Drp1^K^*^38*A*^). Larval locomotor assays were conducted on 10 larvae per genotype per day by quantifying distance crawled in 1 min, as previously described [34]. Assessment of adult locomotion was determined using a negative geotaxis assay. Briefly, age-matched male flies were separated into groups of 10 individuals per genotype that were tested together under the same conditions once a week over 36 days. The proportion of flies climbing to the top of a vertical glass vial (10 cm length, 2.5 cm diameter) over 15 s was determined. For survival assays, male flies from climbing assays were transferred into fresh media every 2–3 days and mortality was scored daily.

### 2.10. Statistical Analysis

All data were exported to Prism 5 (GraphPad Software, Inc.; San Diego, CA, USA) for statistical analysis. For larval and adult locomotor assays and all ER and mitochondrial analyses, statistical significance was determined using one-way ANOVA and Tukey’s multiple comparison post-hoc tests. Life span assays were analysed using the Kaplan Meier log-rank test. For protein expression analysis, statistical significance was determined using a two-tailed *t*-test.

## 3. Results

### 3.1. Mitochondrial Fission Is Disrupted in an Arl6IP1 Knockdown Model of HSP

Previous work in our lab has found that knockdown of the ER-shaping protein Arl6IP1 results in elongation of axonal mitochondria, suggestive of impaired mitochondrial fission [19]. To understand this effect mechanistically, we investigated whether levels of the mitochondrial fission protein Drp1 are disrupted in this model of HSP. Analysis of mRNA expression levels detected no significant change in Drp1 in *Arl6IP1* RNAi *Drosophila* (Figure 1A). Since we were unable to source an antibody which detected *Drosophila* Drp1, we used a *Drp1::FLAG-FlAsH-HA* fusion line to quantify endogenous levels of Drp1 [24]. Analysis of the 82 kDa band detected by HA antibodies, corresponding to full length Drp1, is significantly reduced in *Arl6IP1* RNAi flies compared to controls (Figure 1B).

To restore Drp1 expression in *Arl6IP1* RNAi flies, we used a HA-tagged *UAS-Drp1^wt^* line to increase the expression of Drp1 [25]. Within cells, Drp1 exists as a cytoplasmic pool, with oligomers localised to mitochondria, ER and peroxisomes [14]. Immunostaining of larval motor neurons demonstrates that Drp1 punta localise primarily to mitochondria, though 20–28% of Drp1 does not colocalise with mitochondria and likely represents ER- and peroxisomal-localised Drp1 [14]. On mitochondria, Drp1 puncta were largely localised to areas of constricted mitochondrial staining, likely reflecting an ongoing fission event, or at the ends of mitochondrial tubules, likely marking sites where fission has occurred (Figure 1C,D) [35]. This suggests that overexpressed Drp1 functions to promote mitochondrial fission within motor neurons. Importantly, overexpressed Drp1 marks fission events within Arl6IP1 knockdown lines to a similar level, with no differences in the localisation frequencies of Drp1 puncta within axon bundles as assessed using one-way ANOVA across the three genotypes studied (Figure 1E). 

The ER contacts mitochondria to mark sites for mitochondrial fission [12]. We used electron microscopy to quantify ER–mitochondrial contacts in vivo as confocal microscopy lacks the sufficient resolution to reveal details of the interactions between these organelles which involve membrane apposition of ~10–30 nm (Figure 2) [36]. Ultrastructural analysis revealed that the branch length of ribosome-rich rough ER is unaffected in any of the genotypes studied (Figure 2C). Knockdown of *Arl6IP1* leads to a striking elongation of mitochondria in epidermal cells compared to controls, as indicated by an increase in the average mitochondrial branch length (Figure 2D). To quantify ER–mitochondrial interactions, the percentage of the outer mitochondrial surface in close opposition with the ER (<30 nm) (Figure 2E) and the number of ER–mitochondrial contacts per mitochondrion (Appendix A) was calculated, revealing a significant decrease in the extent of ER–mitochondrial contacts in *Arl6IP1* RNAi larvae compared to controls. We saw no change in the mean length or thickness of ER–mitochondrial contacts (Appendix A). Conversely, overexpression of Drp1 in *Arl6IP1* RNAi larvae blocks mitochondrial elongation and increases the proportion of the mitochondrial membrane in contact with the ER. Together, these findings point to impaired Drp1-mediated mitochondrial fission in an Arl6IP1 knockdown model of HSP.

### 3.2. Increased Drp1 Activity Rescues Mitochondrial Morphology and Neurodegeneration in Arl6IP1 Knockdown Flies

We wanted to validate that increased mitochondrial fission driven by overexpression of Drp1 is mediating the observed rescue in mitochondrial morphology detected in *Arl6IP1* RNAi *Drosophila*. To accomplish this, we generated transgenic flies expressing *Arl6IP1* RNAi in combination with overexpression of a wild type Drp1 (Drp1^WT^) or a dominant negative Drp1 mutant (Drp1^K38A^) and mitochondrial morphology and load within axons of long motor neurons was studied. Consistent with previous results, knockdown of Arl6IP1 results in elongation of axonal mitochondria and reduced mitochondrial load in terminal boutons within posterior neuromuscular junctions (NMJs) [19] (Figure 3A,B). The loss of mitochondrial load from the terminal boutons is not the result of defective mitochondrial trafficking, since no change in mitochondrial flux or trafficking speed was detected in *Arl6IP1* RNAi flies compared to controls (Figure 3C). Instead, we suggest that the reduction in mitochondrial load is due to impaired mitochondrial fission which could limit the production of daughter mitochondria at the synapse. Overexpression of Drp1^WT^ largely rescued axonal mitochondrial morphology and NMJ load (Figure 3A,B). However, overexpression of Drp1^K38A^ failed to rescue, and sometimes enhanced, these mitochondrial phenotypes (Figure 3A,B). This demonstrates that altered mitochondrial morphology in *Arl6IP1* RNAi motor neuron axons can be rescued by increasing mitochondrial fission.

Neuronal knockdown of Arl6IP1 in *Drosophila* results in progressive locomotor deficits recapitulating a key characteristic of HSP [19]. We now show that both larval and adult locomotor defects in *Arl6IP1* RNAi *Drosophila* are largely restored by overexpression of Drp1^WT^ (Figure 4A,B; Appendix A). In contrast, overexpression of Drp1^K38A^ fails to rescue locomotion in *Arl6IP1* RNAi larvae (Figure 4A) and significantly exacerbates the progressive locomotor defect in *Arl6IP1* RNAi flies, particularly beyond 18 days old (Figure 4B; Appendix A). Survival is not consistently altered across the genotypes studied (Figure 4C; Appendix A). Together, these data show that increased mitochondrial fission can help to prevent neurodegeneration in this model of HSP.

### 3.3. Protein Processing Pathways Are Impaired in Arl6IP1 Knockdown Flies Independent of Drp1 Expression

ER-shaping proteins have a highly conserved role in regulating autophagic degradation of the ER, from plants [37] to mammalian cells [38]. Drp1 has also been found in several studies to regulate autophagy, with reduced Drp1 leading to an accumulation of autophagosomes [39], while Drp1 overexpression enhances autophagy [6]. To investigate autophagic degradation in our model of HSP, we used the dual tagged GFP-mCherry-Atg8a marker which labels autophagosomes as yellow (with both GFP and mCherry), while low pH autolysosomes are labelled mostly red due to protonation quenching of GFP [40]. An increase in the red–green ratio of Atg8a puncta therefore acts as a readout of autophagic flux. During insect metamorphosis, several tissues including the fat body undergo precisely timed periods of programmed autophagy. This developmental autophagy obscures detection of starvation-induced autophagy and as a result L2 stage larvae are instead utilised to analyse induced autophagy [41]. Under fed conditions, basal autophagy in late L2 stage larvae is low and Atg8a puncta stain mostly yellow (Figure 5A). Starvation induces autophagy and within control larvae we see an expected marked increase in red Atg8a puncta, indicating the degradative completion of autophagy via the formation of autolysosomes (Figure 5A). However, we find that knockdown of Ar6IP1 impairs this induction of autophagy, with the red–green ratio of Atg8a puncta remaining similar in fed and starved larvae (Figure 5B). Overexpression of Drp1 fails to restore starvation-induced autophagy in *Arl6IP1* RNAi larvae, indicating that this impairment occurs independent of Drp1-mediated mitochondrial fission. 

Impaired autophagy has been reported in several subtypes of HSP [42,43,44] and individuals with mutations in the ER-shaping protein Atlastin 3 (associated with the related disorder hereditary sensory and autonomic neuropathy (HSAN)) have compromised autophagy [45]. However, this is the first-time disrupted autophagy has been reported in an Arl6IP1 model of HSP. To validate this finding, we looked for evidence of impaired protein clearance in our models, as attenuated autophagy has been linked to the accumulation of polyubiquitin aggregates [46,47]. Dissected larval preps were stained with Fk2 antibodies which recognise mono-and poly-ubiquitinylated substrates and free ubiquitin chains (Figure 6A)**.** We found that knockdown of Arl6IP1 results in increased ubiquitin-positive staining within synaptic boutons of distal NMJs of *Arl6IP1* RNAi flies which is not rescued by overexpression of Drp1 (Figure 6B). This confirms that protein degradation is impaired in *Arl6IP1* RNAi flies, independent of Drp1-mediated mitochondrial fission.

## 4. Discussion

The large GTPase Drp1 is a critical regulator of mitochondrial fission, and dysregulation of the mitochondrial fission/fusion balance which has been implicated in the pathogenesis of several neurodegenerative diseases. Here, we study a *Drosophila* model of HSP generated by targeted knockdown of the ER-shaping protein Arl6IP1, mutations in which cause the HSP subtype SPG61 [48]. We report that Drp1 expression is decreased in this HSP model, causing reduced ER–mitochondrial contacts and impaired mitochondrial fission. Overexpression of Drp1 restores mitochondrial fission and partially rescues locomotor deficits, providing evidence that impaired mitochondrial fission contributes to neurodegeneration in this model of HSP. 

Drp1 is a highly dynamic protein which is modulated by a vast array of physiological responses including but not limited to: exercise [49], inflammatory signalling [50] and neuronal differentiation [6]. Dysregulation of Drp1 expression has been reported in samples from patients with Alzheimer’s disease [8] and ALS [11]. Furthermore, Drp1 dysregulation is detected in several models of neurodegeneration, with mutations in the AFG3-like matrix AAA peptidase subunit 2 (AFG3L2) gene, a paralog of the HSP-causing gene SPG7, and knockdown of the ER-shaping protein REEP1 both causing a reduction in the expression of Drp1 protein [21,51]. The mechanism by which Drp1 expression is altered in these models is not known, though REEP1 has been suggested to affect the stability of Drp1 by modulating its phosphorylation status [21]. We now show that Drp1 expression is decreased in an *Arl6IP1* RNAi *Drosophila* model of HSP. This finding emphasises the importance of investigating Drp1-mediated mitochondrial fission in HSP.

There are several mechanisms by which a decrease in Drp1 levels and impaired mitochondrial fission might contribute to neurodegeneration in our model of HSP. One mechanism which must be considered is a disruption in mitophagy, the turnover and maintenance of healthy mitochondria by autophagy. Mitophagy can be regulated by Drp1 activity, with decreased Drp1-mediated mitochondrial fission resulting in hyperfused mitochondria which escape induced mitophagy [52,53]. While this can preserve energy production within a cell in the short term, long-term inhibition of mitophagy results in an accumulation of defective mitochondria contributing to neurodegeneration [54]. Several studies have reported a conserved age-related decline in mitophagy [55,56,57] which would be exacerbated by disorders which cause a reduction in Drp1. Here, we report that Arl6IP1 knockdown flies have impaired induced autophagy which is not restored by overexpression of Drp1. Future work to investigate basal and induced mitophagy in this model of HSP and whether these are impacted by overexpression of Drp1 would help to determine whether this might be contributing to neurodegeneration.

Drp1-mediated mitochondrial fission also regulates how post-mitotic neurons respond to oxidative stress. Reduced Drp1 results in an increase in reactive oxygen species (ROS) and oxidative damage in mouse cerebellar neurons, while treatment with antioxidants reduces mitochondrial elongation and neuronal death [58]. Drp1 may also impact apoptosis in affected neurons however either increased or decreased Drp1 activity has been shown to induce apoptosis in differentiating and cultured neurons [6,59]. Therefore, it remains unclear whether or how this might be contributing to neurodegeneration in different models. 

Finally, lipid biogenesis can be disrupted by alterations to mitochondrial fission/fusion. Within the nervous system, lipid droplets (LDs) play vital roles in energy storage, glia-neuron communication and ROS management and lipid homeostasis is necessary for maintaining neuronal function and synaptic plasticity [60]. Defective mitochondrial fusion causes an accumulation of neutral lipids within lipid droplets in both mitofusin 1 knockout cells and fibroblasts from Charcot-Marie-Tooth disease (CMT) type 2A patients [61,62]. By contrast, loss of the mitofusion 1 ortholog in *Drosophila*, Marf, results in a marked decrease in LDs in an endocrine organ, the ring gland [63]. Most recently, Drp1 was shown to directly regulate LD formation, with targeted knockout of Drp1 resulting in an increase in large LDs within mouse adipose tissue [37]. This LD defect can be rescued by overexpression of Drp1. Disrupted ER, on which LDs form, and decreased Drp1 levels may therefore synergistically contribute to neurodegeneration in our model of HSP by disrupting LD formation. Interestingly, defects in lipid metabolism and storage within LDs are increasingly being associated with HSPs suggesting that lipid biogenesis may contribute to the pathophysiology of these diseases [64].

Given that we have found that restoring Drp1-mediated mitochondrial fission partially rescues locomotor defects in our model of HSP, it may lead us to consider whether Drp1 could be a potential therapeutic target for HSP or other neurodegenerative disorders. Several studies support the suggestion that increased Drp1-mediated mitochondrial fission can have beneficial effects in *Drosophila* models of Alzheimer’s disease, HSP and Parkinson’s disease [21,65]. In fact, even brief overexpression of Drp1 during midlife in *Drosophila* reduces age-associated mitochondrial elongation, facilitates mitophagy and lowers mitochondrial reactive oxygen species (ROS) levels [57]. Moreover, these changes are accompanied by a significant increase in mobility and lifespan, further suggesting that treatments which promote Drp1-mediated mitochondrial fission could be useful to combat degenerative phenotypes. On the other hand, many other studies have emphasised that overexpression of Drp1 in vivo can have harmful effects, significantly disrupting the mitochondrial network and stress pathway induction [66]. Furthermore, inhibition of Drp1 has been shown to be neuroprotective in mammalian cellular and in vivo models of HSP and Parkinson’s disease [59,67]. It is therefore likely that long-term inhibition or activation of mitochondrial fission would not be a suitable therapeutic approach due to potential disruption to mitophagy, apoptosis, or lipid biosynthesis. Instead, an approach whereby a healthy balance between mitochondrial fission and fusion is restored would be preferable.

## Figures and Tables

**Figure 1 brainsci-10-00646-f001:**
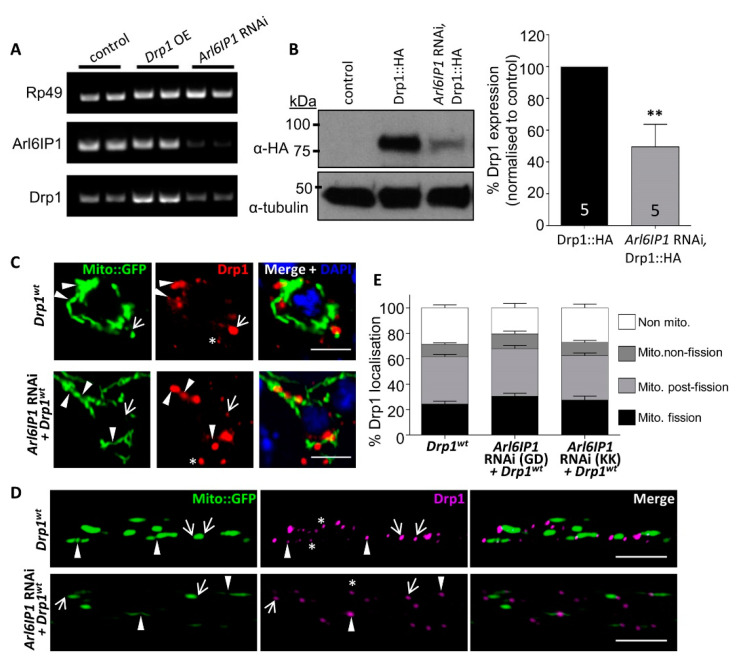
Decreased levels of the mitochondrial fission protein Drp1 in *Arl6IP1* RNAi *Drosophila*. (**A**) Polymerase chain reaction (PCR) amplification of Rp49, Arl6IP1 and Drp1 cDNA from progeny of *w*^1118^ (control), UAS-*Drp1* (*Drp1* OE) or UAS-*Arl6IP1* RNAi (GD) (*Arl6IP1* RNAi) crossed to *da-GAL4* flies. (**B**) Western blot analysis of protein lysates from progeny of FLAG-FlAsH-HA Drp1 fusion (Drp1::HA) or *Arl6IP1* RNAi (GD) with FLAG-FlAsH-HA Drp1 (*Arl6IP1* RNAi, Drp1::HA) crossed to *da-GAL4* flies. Data are expressed as means ± SEM (*n* = 5 independent experiments), and values significantly different from control were determined by two-tailed *t*-test (**, *p* < 0.01). (**C**,**D**) Representative confocal images of L3 larvae showing Drp1 and mitochondria in motor neuron cell bodies (**C**) and axons (**D**). Larvae are progeny of *ok6-GAL4, mitoGFP* crossed to *UAS-Drp1^wt^HA* (*Drp1^wt^*) or UAS-*Arl6IP1* RNAi (GD), UAS-*Drp1^wt^HA* (*Arl6IP1* RNAi + *Drp1^wt^*) flies. Drp1 puncta co-localising with mitochondria were primarily detected at areas of reduced mitochondrial staining (arrowheads; likely during a fission event) or to the end of mitochondrial tubules (arrows; likely post-fission), while a subset do not colocalise with mitochondrial staining (asterix). (**E**) Quantifications of the proportion of Drp1 puncta within axon bundles localised to points of: mitochondrial constriction (Mito. Fission), ends of mitochondrial tubules (Mito. post-fission), mitochondria generally (Mito. non-fission), and independent of mitochondria (Non mito,) (*n* = 6–18 larvae). Scale bars = 5 μm.

**Figure 2 brainsci-10-00646-f002:**
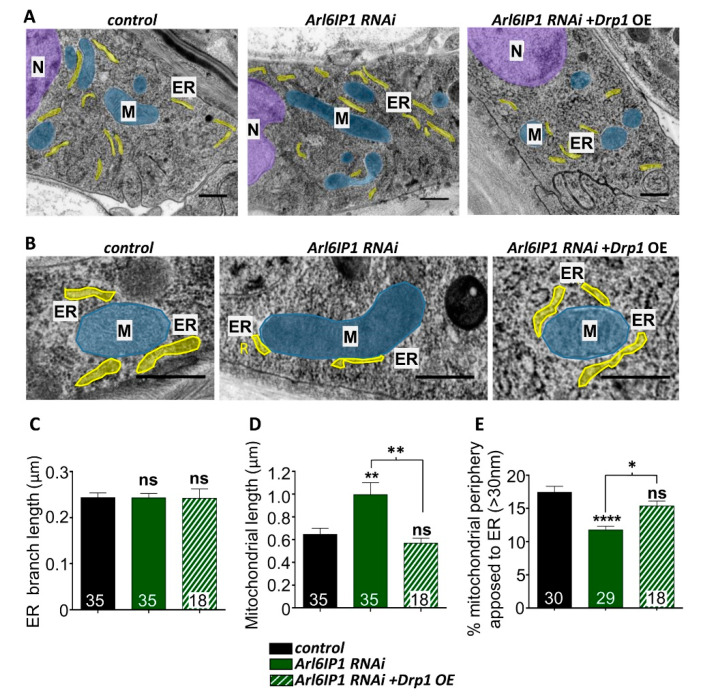
Overexpression of Drp1 increases ER–mitochondrial contacts and blocks mitochondrial elongation in *Arl6IP1* knockdown *Drosophila*. (**A**) Representative electron micrographs revealing mitochondrial (blue; M) and ER (yellow; ER) structures within larval epidermal cells. These cells were chosen as they allowed reproducible visualisation of cytoplasmic components surrounding the nucleus (purple; N). Larvae are progeny of *w*^1118^ (control), UAS-*Arl6IP1* RNAi (GD) or UAS-*Arl6IP1* RNAi (GD)*. UAS-Drp1* (*Arl6IP1* RNAi + *Drp1* OE) flies crossed to *da-GAL4*. (**B**) Representative electron micrographs of ER–mitochondrial contacts in the three genotypes studied. Quantification of ER branch length (**C**), mitochondrial length (**D**) and proportion of the mitochondrial membrane contacting ER (**E**) in the three genotypes studied. Data are expressed as means ± SEM (*n* = 18–35 cells from three independent experiments) and values significantly different from control were determined by one-way ANOVA and Tukey’s multiple comparisons test (*, *p* < 0.05; **, *p* < 0.01; ****, *p* < 0.0001; ns, *p* > 0.05). All scale bars = 5 nm.

**Figure 3 brainsci-10-00646-f003:**
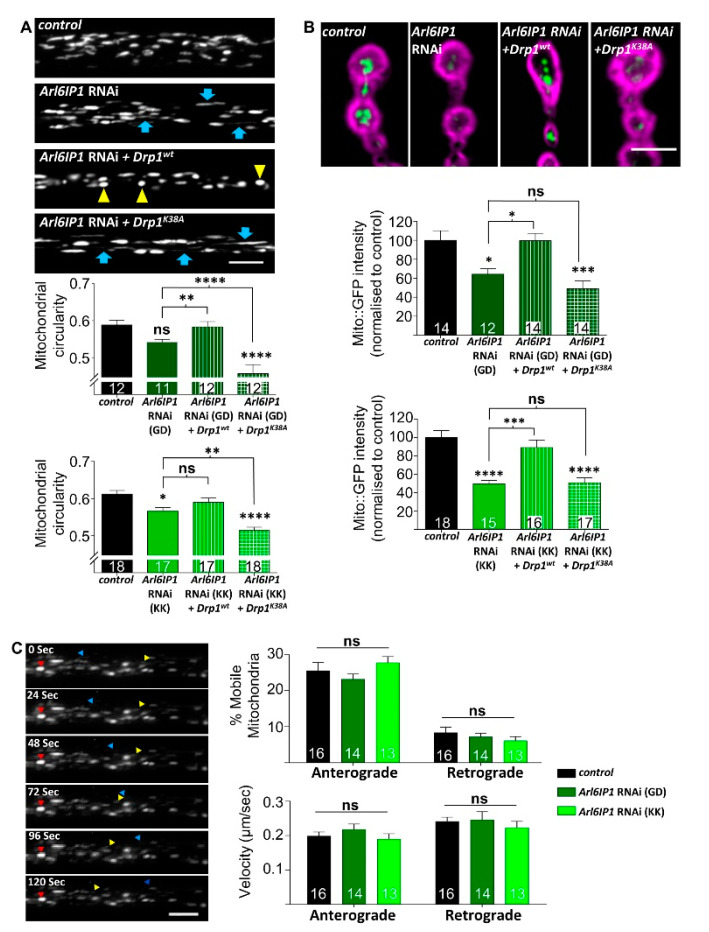
Increased mitochondrial fission via overexpression of *Drp1^wt^* restores mitochondrial morphology and localisation within distal portions of *Arl6IP1* RNAi long motor neurons. Larvae are progeny of *ok6-GAL4* flies carrying mito::GFP crossed to *w*^1118^ (control), UAS-*Arl6IP1* RNAi (*Arl6IP1* RNAi), UAS-*Arl6IP1* RNAi*.UAS-Drp1^WT^* (*Arl6IP1* RNAi + *Drp1^WT^*) or UAS-*Arl6IP1* RNAi*.UAS-Drp1^K^*^38*A*^ (*Arl6IP1* RNAi + *Drp1^K^*^38*A*^)*. w*^1118^ and *UAS-Al6IP1-RNAi* lines from two independent RNAi stocks (GD and KK) were tested separately. (**A**) Representative confocal images showing motor neuron mitochondria in posterior axon bundles. Overexpression of *Drp1^WT^* in *Arl6IP1* RNAi larvae reduced mitochondrial elongation (yellow arrowheads) while overexpression of *Drp1^K^*^38*A*^ further increases mitochondrial elongation (blue arrows) compared to *Arl6IP1* RNAi alone. Graphs show mitochondrial circularity. (**B**) Representative confocal images showing mitochondria (green) within terminal synaptic boutons (DLG, magenta) of posterior NMJs. Graphs show mito::GFP intensity within posterior boutons normalised to controls. Live axons expressing mito::GFP were imaged at 2-s intervals for 2 min with representative stills taken every 24 s over the 2-min video shown in (**C**). Mitochondria were categorised as stationary (red arrowhead), oscillatory or moving (anterograde—blue arrowhead, retrograde—yellow arrowhead). Graphs show proportion of moving mitochondria and mitochondrial velocity within *Arl6IP1* RNAi motor neuron axons in both anterograde and retrograde direction. All data are expressed as means ± SEM (*n* = 11–18 larvae from three independent experiments) and values significantly different from controls were determined by one-way ANOVA and Tukey’s multiple comparisons test (*, *p* < 0.05; **, *p* < 0.01; ***, *p* < 0.001 ****, *p* < 0.0001; ns, *p* > 0.05). Scale bars = 5 μm.

**Figure 4 brainsci-10-00646-f004:**
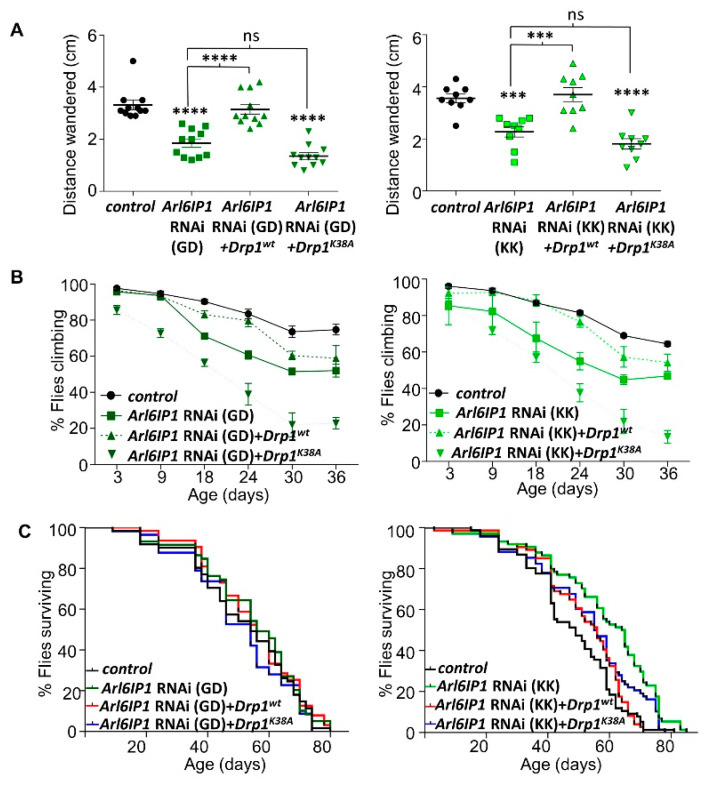
Increased mitochondrial fission induced by overexpression of *Drp1^wt^* partially rescues locomotor deficits in *Arl6IP1* RNAi *Drosophila*. All *Drosophila* are progeny of *nSyb-GAL4* flies crossed to *w*^1118^ (control), UAS-*Arl6IP1* RNAi (*Arl6IP1* RNAi), UAS-*Arl6IP1* RNAi*.UAS-Drp1^WT^* (*Arl6IP1* RNAi + *Drp1^WT^*) or UAS-*Arl6IP1* RNAi*.UAS-Drp1^K^*^38*A*^ (*Arl6IP1* RNAi + *Drp1^K^*^38*A*^)*. w*^1118^ and *UAS-Al6IP1-RNAi* lines from two independent RNAi stocks (GD and KK) were tested separately. (**A**–**C**) Graphs show distance crawled by larvae in 1 min (**A**; *n* = 9–10 experiments; 10 larvae per experiment; and values significantly different from controls were determined by one-way ANOVA and Tukey’s multiple comparisons test (***, *p* < 0.001 ****, *p* < 0.0001; ns, *p* > 0.05)**)**, percent flies climbing in a negative geotaxis assay (**B**; *n* = 11–27 experiments per genotypes; see Appendix A for statistical analysis) and lifespan assays (**C**; *n* = 57–76 flies per genotype; see Appendix A for statistical analysis).

**Figure 5 brainsci-10-00646-f005:**
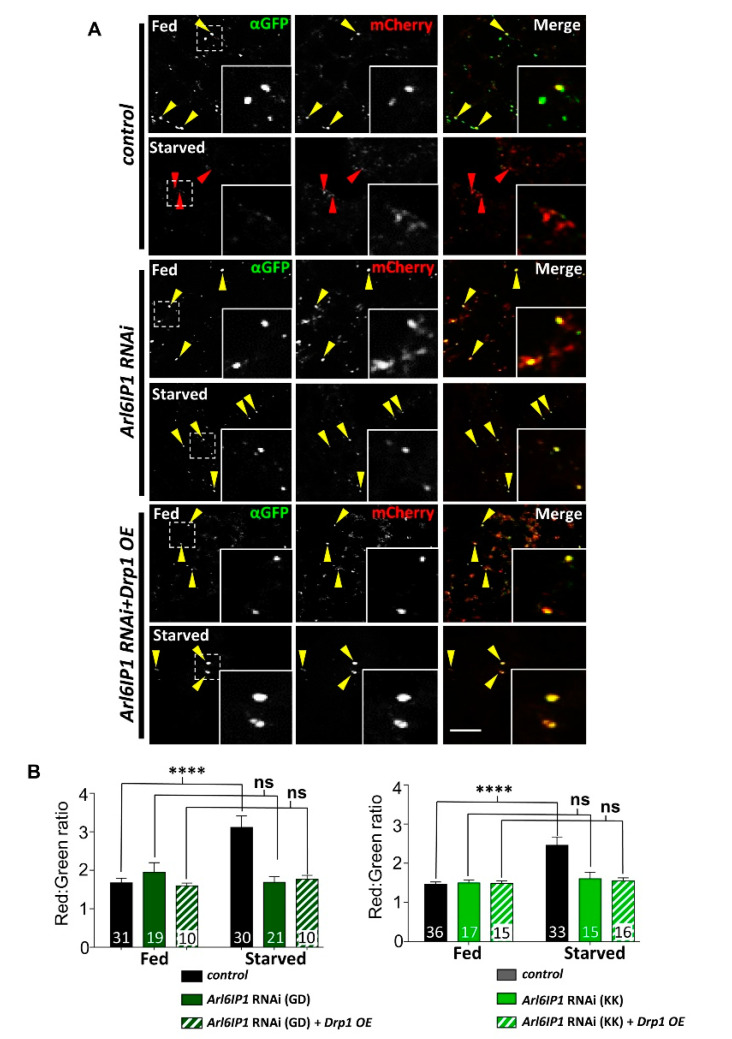
Knockdown of Arl6IP1 RNAi in impaired included autophagy which is not affected by overexpression of Drp1. (**A**) Representative confocal sections showing *Drosophila* fat body cells expressing the autophagosomal maker mCherry-GFP-Atg8a from late L2 larvae under fed or starved conditions. Larvae are progeny of *Cg-GAL4*. UAS-GFP-mCherry-Atg8a flies crossed to *w*^118^ (controls), UAS-*Arl6IP1* RNAi (*Arl6IP1* RNAi) or UAS-*Arl6IP1* RNAi*.UAS-Drp1* (*Arl6IP1* RNAi + *Drp1* OE). At cytosolic pH both the GFP and mCherry tags are detectible (yellow arrowheads) while within the low pH of an autolysosome the GFP is quenched and only mCherry is detectible (red arrowheads). Scale bar = 10 μm. (**B**) Quantification of mean red (mCherry) and green (GFP) intensities of Atg8a-tagged puncta under fed and starved condition in *UAS-Al6IP1-RNAi* lines from two independent RNAi stocks (GD and KK) which were tested separately. Data are expressed as means ± SEM (*n* = 10–36 larvae) and values significantly different from control were determined by one-way ANOVA and Tukey’s multiple comparisons test (****, *p* < 0.0001; ns, *p* > 0.05).

**Figure 6 brainsci-10-00646-f006:**
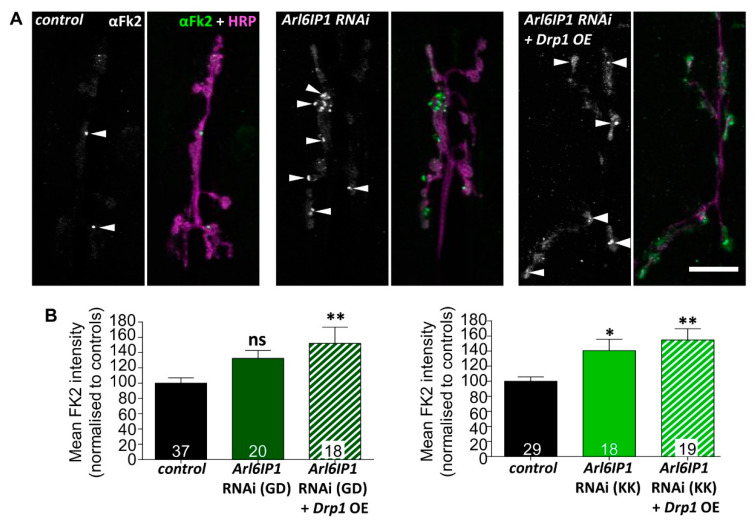
Knockdown of *Arl6IP1* causes increased aggregation of ubiquitinated proteins in neurons which is not rescued by Drp1 overexpression. (**A**) Representative single confocal sections showing Fk2-positive puncta accumulation (green; white arrowheads) within posterior axons (HRP; magenta). Larvae are progeny of *da-GAL4* flies crossed to *w*^1118^ (controls), UAS-*Arl6IP1* RNAi (*Arl6IP1* RNAi) or UAS-*Arl6IP1* RNAi*.UAS-Drp1* (*Arl6IP1* RNAi + *Drp1* OE). Scale bar = 20 μm. (**B**) Quantification reveals increased Fk2-positive staining in NMJ boutons of flies expressing *Arl6IP1* RNAi from two independent RNAi stocks (GD and KK, tested separately) compared to relevant controls. Data are expressed as means ± SEM (*n* = 18–37 larvae from three independent experiments) and values significantly different from control were determined by one-way ANOVA and Tukey’s multiple comparisons test (*, *p* < 0.05; **, *p* < 0.01; ns, *p* > 0.05).

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
