# Peer review of "Loss of the Mitochondrial Fission GTPase Drp1 Contributes to Neurodegeneration in a Drosophila Model of Hereditary Spastic Paraplegia"

_brainsci, 2020, doi:10.3390/brainsci10090646_

Round 1

Reviewer 1 Report

This study by Fowler et al. nicely demonstrates molecular phenotypes in an Arl6IP1 knockdown fly model of HSP. They show that Drp1 level in decreased in these flies, resulting in elongated mitochondria. Overexpression of Drp1 but not its dominant negative form rescues the phenotypes that are related to decreased mitochondrial fission. Independent of Drp1, Arl6IP1 knockdown impairs autophagy flux, resulting in accumulation of ubiquitinated proteins. The manuscript is clearly written.

I have a few questions/comments:
- Regarding the result of ER-mitochondria contacts decreasing in Arl6IP1 knockdown flies, it is presented as proportion of the mitochondrial membrane
contacting ER. As at the same time, mitochondrial length is increased, is this the reason for having a smaller proportion of mitochondrial membrane in contact with ER? Or are there really overall less contacts between mitochondria and ER? This should be clarified, or the current conclusion of ‘reduced ER-mitochondrial contacts’ modified.
-Authors mention that the loss of mitochondrial load from the terminal boutons is not the result of defective mitochondrial trafficking, however, it is not clear what then is the cause of reduced mitochondrial load. This should be elaborated.
- Starting from Fig 3, it could be clarified in Figure legends what GD and KK are.

Author Response

Thank you for your constructive comments which we respond to sequentially as follows:

  1. Physiological changes can alter the number, length or thickness of ER-mitochondrial contacts and in many ways this can best be represented by analysis of the proportion of mitochondrial perimeter in contact with the ER (Giacomello and Pellegrini, 2916). However, given the increase in mitochondrial length (and therefore perimeter area) that we observe in our model, it fair to question whether this alone could be driving the reduction in the proportion of mitochondria in contact with the ER. We have therefore provided additional information detailing the number, length and thickness of ER-mitochondrial contacts in our models (Supplemental Figure 1). These data support our conclusion that knockdown of Arl6IP1 reduces ER-mitochondrial contacts in Drosophila.
  2. The likely cause of reduced mitochondrial load at motor neuron synapses has been elaborated in the text as follows: “we suggest that the reduction in mitochondrial load is due to impaired mitochondrial fission which could limit the production of daughter mitochondria at the synapse.
  3. Further clarification about the RNAi library lines used has been provided in the methods section:

Both libraries were constructed by inserting inverted repeats of known genes into Drosophila, with the key difference being that GD library insertions are P-element based transgenes with random insertion sites, whereas the KK library contains phiC31-based transgenes with a single, defined insertion site.”

as well as in the figure legends for Figure 3 onwards e.g. legend for Figure 3:

w1118 and UAS-Al6IP1-RNAi lines from two independent RNAi stocks (GD and KK) were tested separately.” and e.g. legend for Figure 5: “Quantification of mean red (mCherry) and green (GFP) intensities of Atg8a-tagged puncta under fed and starved condition in UAS-Al6IP1-RNAi lines from two independent RNAi stocks (GD and KK) which were tested separately.

Reviewer 2 Report

Loss of the mitochondrial fission GTPase Drp1 contributes to neurodegeneration in a Drosophila model of hereditary spastic paraplegia #brainsci-926921

Drp1 protein is known to be a key regulator of mitochondrial fission, being recruited at regions where mitochondrial membrane contacts the endoplasmic reticulum (ER). While previous data indicate that Drp1, ER-mitochondria contacts and ER morphology play important roles in neurodegeneration, there is only limited information about how mutations in ER-shaping proteins cause neurodegenerative diseases such as hereditary spastic paraplegia (HSP). Here, authors hypothesize that in HSP, the disruption of axonal ER morphology can lead to neuronal degeneration by Drp1-mediated impairing of mitochondrial morphology. Using Arl6IP1 RNAi expression in Drosophila as an HSP model, authors show that Arl6IP1 is required for Drp1 expression, and also that Drp1 overexpression can rescue the mitochondrial elongation defects associated with Arl6IP1 knocked-down animals. On the other hand, the locomotor defects associated with Arl6IP1 RNAi expression are only partially rescued by Drp1 overexpression, suggesting that neurodegeneration is not only due to the mitochondrial fission defects. Interestingly, authors convincingly show that Arl6IP1 RNAi impair autophagy, and that this is independent of Drp1. Since autophagy misfunction related with neurodegeneration, this might suggest another mechanism by which Arl6IP1 loss produces HSP. However, this still remains to be properly studied.

Given the high structural and functional evolutionary conservation of both ER and mitochondrial morphology, and their importance in neurodegeneration, this work should be of wide interest, validating Drosophila as a useful model for further studies in the field of HSP. In general, authors provide solid data and a good experimental design to test their hypothesis. However, I found a number of points in the manuscript that might be improved or that I would recommend addressing to fully support the conclusions reached:

Major comments

  1. Authors analysed late L2 stage larvae for autophagy analysis (Page 11, Line 328), and for electron microscopy they used early L2 stage larvae (Page 4, Line 167). Since authors used L3 stage for all the other experiments performed in larvae, could they please justify the use of L2 (instead of L3) in both autophagy and electron microscopy experiments?
  2. According reference 19 (Page 15, Line 512), expression of Arl6IP1 RNAis drastically reduces Arl6IP1 mRNA levels, but not completely. In order to fully understand the requirement of Arl6IP1 protein in mitochondrial fission, it would be interesting to see the effect of the complete loss of function of Arl6IP1. In the absence of a null mutant allele, did the authors try to express both RNAi lines simultaneously, or to enhance the RNAis efficiency (e.g. by co-expressing UAS-Dicer)?
  3. According the EM data presented in Figure 2, I would suggest that ER-mitochondria contact sites are not affected in Arl6IP1 knocked-down animals, but only the size of the mitochondria. Therefore, between control and Arl6IP1 RNAi (rescued in Arl6IP1 RNAi + Drp1 OE), it might only change the proportion of the mitochondrial surface that it is in contact with the ER. I found the authors described this properly in the main text of the Results section. However, in the Abstract (Page 1, Lines 20 and 22), the Introduction (Page 2, Line 71), the Discussion (Page 13, Line 375) and in the title of Figure 2 legend (Page 7, Line 256), I found misleading or not fully supported by the data the conclusion reached by the authors: ¨reduced (in Arl6IP1 RNAi) or restored (in Arl6IP1 RNAi + Drp1 OE) ER-mitochondrial contacts¨. I would recommend to modify this conclusion in order to be supported by the data presented.

Minor comments

- Figures 1-6 look a little bit blurry, not only the images but also the labels. This might be due to a loss of quality when exporting from the original Figure files. If possible, I would recommend to improve the quality of the Figures.

- Figures 1, 2, 3, 5 and 6: I would find the quantifications more informative if datapoints were plotted in the bar graphs (as they are in Figure 4A).

- Page 2, Line 59: Since it is known that some ER-shaping proteins perform additional functions to their role in shaping ER, I would suggest to omit ¨whose normal function¨, and instead only indicate that they regulate the organization of the ER network.

- Page 2, Line 85: Although authors describe that ¨unless otherwise indicated, phenotypes were analysed in both RNAi lines¨, all Figure legends in the article should include information about which specific RNAi line was used for each of the representative images presented.

- Page 5, Line 208: Authors conclude that ¨Analysis of mRNA expression levels detected a slight reduction in Drp1 in Arl6IP1 RNAi Drosophila¨. However, in Figure 1A, I find difficult to detect significant differences for Drp1 cDNA levels between Control and Arl6IP1 RNAi conditions. Therefore, I would recommend to include a quantification of the intensity levels of the shown bands (as it is done in Figure 1B). Otherwise, I would recommend to conclude that the expression of Arl6IP1 RNAi does not produce any significant change in the mRNA levels of Drp1.

- Page 9, Line 296: The time indicated between recordings, ¨24s¨, is different than the one indicated in the Material and Methods section, ¨2s¨ (Page 4, Line 158). Please correct the one that is wrong.

Author Response

Thank you for your constructive comments which we respond to sequentially as follows:

  1. Wandering L3 larvae are the standard stage analysed for NMJ morphology and function and therefore this stage has been utilised for most of the experiments in this study.

For the autophagy study we used late L2 larvae because “During insect metamorphosis, several tissues including the fat body undergo precisely timed periods of programmed autophagy. This developmental autophagy obscures detection of starvation-induced autophagy and as a result late L2 or early L3 stage larvae are utilised to analyse induced autophagy rather than wandering L3 larvae (Mauvezin et al., 2014).” This text has been added to the results section (page 12) to clarify this.

For the EM study, early L2 stage larvae were used as these were the appropriate size for the high pressure freezing unit that our imaging facility possesses. This has been explained in the manuscript text.

  1. Our lab has generated Arl6IP1 KO flies using CRISPR-Cas9 gene editing which are the subject of another manuscript in preparation. We observe similar behavioural and molecular changes in the KO and RNAi knockdown models indicating that the RNAi models are a good indicator of Arl6IP1 function in vivo.
  2. Similar to comment 1 by reviewer 1, we have therefore provided additional information detailing the number, length and thickness of ER-mitochondrial contacts in our models (Supplemental Figure 1). These data support our conclusion that knockdown of Arl6IP1 reduces ER-mitochondrial contacts in Drosophila.
  3. Low resolution PDF figures were uploaded for review in error (our apologies for this). High resolution JPEGs have since been uploaded.
  4. With so many data points for each genotype for most of the graphs presented in this paper (30+ in most cases), we believe that bar charts representing mean +/- SEM is the clearest way to present these data.
  5. The sentence has been adjusted as requested.
  6. Similar to comment 3 by reviewer 1, we have addressed the lack of clarity about the RNAi library lines used by including much more detail within both the methods section and figure legends.
  7. We routinely detected 10-15% reduction in Drp1 mRNA in Arl6IP1 RNAi flies. However, given that this analysis was conducted using semi-quantitative PCR we would not be confident to include quantification of this data in the published manuscript. We have therefore altered the text as suggested to read “Analysis of mRNA expression levels detected no significant change in Drp1 in Arl6IP1 RNAi Drosophila”.
  8. There is not an error here however, we have clarified the description in the figure legend to read “Live axons expressing mito::GFP were imaged at 2 second intervals for 2 minutes, with representative stills taken every 24 seconds over the 2 minute video shown in (C).”